# Exact results for a fractional derivative of elementary functions

Gavriil Shchedrin⋆, Nathanael C. Smith, Anastasia Gladkina and Lincoln D. Carr

Colorado School of Mines, Golden, Colorado 80401, USA

⋆ shchedrin@mines.edu

## Abstract

We present exact analytical results for the Caputo fractional derivative of a wide class of elementary functions, including trigonometric and inverse trigonometric, hyperbolic and inverse hyperbolic, Gaussian, quartic Gaussian, Lorentzian, and shifted polynomial functions. These results are especially important for multi-scale physical systems, such as porous materials, disordered media, and turbulent fluids, in which transport is described by fractional partial differential equations. The exact results for the Caputo fractional derivative are obtained from a single generalized Euler's integral transform of the generalized hypergeometric function with a power-law argument. We present a proof of the generalized Euler's integral transform and directly apply it to the exact evaluation of the Caputo fractional derivative of a broad spectrum of functions, provided that these functions can be expressed in terms of a generalized hypergeometric function with a power-law argument. We determine that the Caputo fractional derivative of elementary functions is given by the generalized hypergeometric function. Moreover, we show that in the most general case the final result cannot be reduced to elementary functions, in contrast to both the Liouville–Caputo and Fourier fractional derivatives. However, we establish that in the infinite limit of the argument of elementary functions, all three definitions of a fractional derivative - the Caputo, Liouville–Caputo, and Fourier - converge to the same result given by the elementary functions. Finally, we prove the equivalence between Liouville–Caputo and Fourier fractional derivatives.

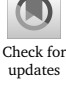

## Contents

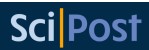

## 1   Introduction

The notion of a fractional derivative and fractional integral of any order, real or complex, is a profound concept in calculus, complex analysis, and the theory of integro-differential equations [1, 2]. These fractional operators have unique mathematical properties and remarkable relations to special functions and integral transforms [3–6]. Apart from its applications in pure mathematics and mathematical physics, the notion of a fractional derivative has found a number of applications in fundamental and applied physics [7]. Indeed, transport phenomena in a wide class of multi-scale physical systems, such as porous materials, disordered media, and turbulent fluids are described by the fractional diffusion equation [8–10]. Only within the framework of fractional partial differential equations can the properties of multi-scale physical systems, such as non-Gaussian statistics, non-Fickian transport, non-locality, fractional geometry, and long range correlations, be taken into account in a simple, unified, and systematic way [7, 11]. The optimal transport through the living porous systems, such as animal tissues and leaves that are made of a highly sophisticated hierarchical network of pores and tubes, is governed by the Murray's law [12–18]. Recently, Murray's law was successfully used to design synthetic materials, which allow one to achieve an enhanced transfer rate and mass exchange with applications that range from fast gas detection sensors to highly efficient electrical batteries [19]. Furthermore, it was shown that living systems, such as neural clusters and heart cell arrays exhibit multiple time scales of adaptation, which, in turn, are governed by a fractional derivative of slowly varying stimulus parameters [20, 21].

The progress in both fundamental and mathematical physics is heavily influenced by integrable models, such as the hydrogen atom and harmonic oscillator, the evolution of which is governed by partial differential equations (PDEs). Naturally, exact results in fractional PDEs [22–25] give deep insight into physics that govern systems characterized by multiple spatial and temporal scales. The central object in fractional PDEs is a fractional derivative, which can be defined in various ways [1, 7]. Among the multiplex of fractional derivatives, the Caputo fractional derivative [1, 7, 26] has been proven the most effective in physical applications [7]. Powerful exact methods and numerical techniques were developed that allowed one to evaluate fractional derivatives of a wide class of functions [1–3, 6, 22–25, 27–40]. However, these methods did not provide a *single and universal* method that could be used in finding exact expressions for the Caputo fractional derivative of elementary functions, such as the Gaussian, Lorentzian, trigonometric and hyperbolic functions, which play a paramount role in physical applications [7]. In this paper, we construct a single method based on the generalized Euler's

integral transform (EIT) that enables the exact evaluation of the Caputo fractional derivative of a broad spectrum of elementary functions, provided that these functions can be expressed in terms of the generalized hypergeometric function with a power-law argument. We compare the obtained results for the Caputo fractional derivative with the Liouville–Caputo and Fourier fractional derivatives. Specifically, we show that the Caputo fractional derivative of elementary functions is given in terms of the generalized hypergeometric function, which in the most general case, cannot be reduced to elementary functions, in contrast to both the Liouville–Caputo and Fourier fractional derivatives. However, we find that in the infinite limit of the argument of elementary functions all three definitions of a fractional derivative - the Caputo, Liouville–Caputo, and Fourier - converge to the same result. Moreover, we establish the complete equivalence between the Liouville–Caputo and Fourier fractional derivative, despite the fact that the latter derivative is defined in the momentum space while the former derivative is defined in the configuration space.

The rest of this paper has the following organization. In section **2** we introduce the main idea that allows us to translate the Caputo fractional derivative into the generalized EIT. In section **3** we present the proof of the generalized EIT. The consecutive sections **4** to **8** present direct implementation of the generalized EIT for the specific case of the Caputo fractional derivative of trigonometric and inverse trigonometric, hyperbolic and inverse hyperbolic, Gaussian, quartic Gaussian, Lorentzian, and shifted polynomial function, correspondingly. Section **9** introduces the Liouville–Caputo and Fourier fractional derivatives and shows the complete equivalence between them. Finally, in section **10** we present the correspondence between the Caputo, Liouville–Caputo, and Fourier fractional derivatives.

## 2   The main idea in a nutshell

In this section we define the Caputo fractional derivative and formulate it in terms of the generalized EIT. This transform formulates a definite integral of the beta-type distribution multiplied by the hypergeometric function with polynomial argument in terms of a single hypergeometric function of a higher order. The transformation of an elementary function into the generalized hypergeometric function enables us to formulate the Caputo fractional derivative in terms of the generalized EIT. The EIT effectively transforms the Caputo fractional derivative into a system of linear equations, which can be readily solved. Thus, we obtain an exact analytical result for the Caputo fractional derivative of a wide class of elementary functions, provided that they can be expressed in terms of the generalized hypergeometric function with a polynomial argument.

The Caputo fractional derivative of a fractional order $0 < \alpha < 1$ is defined as [1, 2]

$$^{C}D_{x}^{\alpha}f(x) = \frac{1}{\Gamma(1-\alpha)} \int_{0}^{x} dt \, (x-t)^{-\alpha} \frac{df(t)}{dt}. \tag{1}$$

Strictly speaking, the Caputo fractional derivative is given only for non-integer values of the fractional order, i.e., $\alpha \notin \mathbb{N}$ [1, 2]. In the special case of the integer values of the parameter $\alpha = n \in \mathbb{N}$, the Caputo fractional derivative is *defined* in terms of the integer order derivative of the $n^{\text{th}}$ order. We will find that except for the special case $\alpha = 0$ the definition of the Caputo fractional derivative given by Eq.(1) for integer values of $\alpha = n \in \mathbb{N}$ is in fact consistent with the integer $n^{\text{th}}$ order derivative. However, in the case of $\alpha = 0$, the Caputo fractional derivative given by Eq.(1) results in a shift of the function by its value at the origin, i.e., $^{C}D_{x}^{\alpha=0}f(x) = f(x) - f(0)$. Based on multiple examples of elementary functions ranging from trigonometric and inverse trigonometric functions to Gaussian and Lorentzian functions, we find that the Caputo fractional derivative of the order $0 \leq \alpha \leq 1$ *continuously* deforms the curve

given by the Caputo fractional derivative of the $\alpha = 0^{\text{th}}$ order into $\alpha = 1^{\text{st}}$ order derivative. However, except for the special case $f(0) = 0$, the requirement ${}^{\text{C}}D_x^{\alpha=0}f(x) \equiv f(x)$ results in a discontinuous jump for the Caputo fractional derivative in the vicinity of $\alpha = 0$. In order to avoid such a discontinuity, we impose the definition given by Eq.(1) for all $0 \le \alpha \le 1$.

First, we notice that any elementary function and its derivative can be expressed in terms of the generalized hypergeometric function ${}_AF_B$, e.g. [41, 42],

$$
\begin{aligned}
\sin(x) &= x\; {}_0F_1\big[\,; 3/2; -x^2/4\big], \\
\exp(-x^2) &= {}_1F_1\big[1; 1; -x^2\big].
\end{aligned}
\tag{2}
$$

Next, we perform a re-scaling of the argument, $t \to xt$, in Eq.(1), and express an elementary function $f(x)$ in terms of a generalized hypergeometric function. Thus, we can represent the Caputo fractional derivative in Eq.(1) of an elementary function $f(x)$ in terms of the integral transform,

$$
{}^{\text{C}}D_x^{\alpha}f(x) = g(\alpha, x) \int_0^1 dt\; t^{c-1}(1-t)^{d-c-1}\; {}_AF_B\left[\begin{array}{c} a_1, \ldots, a_A \\ b_1, \ldots, b_B \end{array}; zt^m\right],
\tag{3}
$$

where the arrays of constants $(a_1 \cdots a_A) \equiv \vec{a}$ and $(b_1 \cdots b_B) \equiv \vec{b}$ in the argument of the generalized hypergeometric function ${}_AF_B(\vec{a}, \vec{b}; zt^m)$, along with constants $c$, $d$, $m$, and functions $g(\alpha, x)$ and $z = z(x)$ depend on the specific choice of the function $f(x)$, fractional order $\alpha$, and argument $x$. In this paper we will restrict our choice to the integer powers of the argument of the hypergeometric function, i.e., $m \in \mathbb{N}$. The integral representation given by Eq.(3) of the Caputo fractional derivative originally defined by Eq.(1) is nothing but the generalized EIT, as compared to the conventional EIT, given by [43–45],

$$
{}_{A+1}F_{B+1}\left[\begin{array}{c} a_1, \ldots, a_A, c \\ b_1, \ldots, b_B, d \end{array}; z\right] = \frac{\Gamma(d)}{\Gamma(c)\Gamma(d-c)} \int_0^1 dt\; t^{c-1}(1-t)^{d-c-1}\; {}_AF_B\left[\begin{array}{c} a_1, \ldots, a_A \\ b_1, \ldots, b_B \end{array}; zt\right].
\tag{4}
$$

We shall note, however, that the Caputo fractional derivative of elementary functions involves the generalized hypergeometric function with a *power-law* argument in contrast to the conventional EIT given by Eq.(4), which is formulated in terms of the generalized hypergeometric function with a *linear* argument. In the next section we prove that the generalized EIT for the hypergeometric function with a power-law argument is given in terms of a single hypergeometric function of a higher order. This will allow us to obtain exact analytical results for the Caputo fractional derivative of a broad class of elementary functions, including, but not limited to, trigonometric and inverse trigonometric, hyperbolic and inverse hyperbolic, Gaussian, quartic Gaussian, and Lorentzian functions. Even though we restricted the order of the Caputo fractional derivative to be $0 \le \alpha \le 1$, the obtained results could be easily extended to a general case $0 \le \alpha < \infty$ by employing the semi-group property of a fractional derivative [1–3]. The semi-group property for the Caputo fractional derivative of a factional order $0 \le \alpha \le 1$ can be directly established by expressing it in terms of the Riemann–Liouville fractional derivative,

$$
{}^{\text{C}}D_x^{\alpha}[f(x)] = {}^{\text{RL}}D_x^{\alpha}[f(x) - f(0)].
\tag{5}
$$

Thus, the semi-group property of the Caputo derivative follows directly from the semi-group property of the Riemann–Liouville derivative [2].

## 3 Generalized Euler's integral transform of the hypergeometric function with a power-law argument

In the previous section we have transformed the Caputo fractional derivative into the generalized EIT of the hypergeometric function with a power-law argument. The goal of this section

is to derive the generalized EIT of the hypergeometric function with a power-law argument in terms of the hypergeometric function of a higher order. Specifically, we will prove the following result that holds for the generalized EIT,

$$
{}_{A+m}F_{B+m}\left[\begin{array}{cc} a_1,\ldots,a_A, & c_0,\cdots c_{m-1} \\ b_1,\ldots,b_B, & d_0,\cdots d_{m-1} \end{array};z\right]
$$
$$
= \frac{\Gamma(d)}{\Gamma(d-c)\Gamma(c)}\int_0^1 dt\; t^{c-1}(1-t)^{d-c-1}\; {}_AF_B\left[\begin{array}{c} a_1,\ldots,a_A \\ b_1,\ldots,b_B \end{array};zt^m\right],
\tag{6}
$$

where the constants $c_j$ and $d_j$ are given by $c_j = (c+j)/m$, and $d_j = (d+j)/m$ with index $j$ spanning $j \in [0,1,\cdots,m-1]$.

We begin the proof of Eq.(6) by considering the integral,

$$
{}_AJ_B = \int_0^1 dt\; t^{c-1}(1-t)^{d-c-1}\; {}_AF_B\left[\begin{array}{c} a_1,\ldots,a_A \\ b_1,\ldots,b_B \end{array};zt^m\right].
\tag{7}
$$

First, we expand the hypergeometric function in the hypergeometric series,

$$
{}_AJ_B = \sum_{n=0}^{\infty}\frac{(a_1)_n\cdots(a_A)_n}{(b_1)_n\cdots(b_B)_n}\frac{z^n}{n!}\int_0^1 dt\; t^{c-1}(1-t)^{d-c-1}t^{mn},
\tag{8}
$$

where $(a)_n$ is the Pochhammer symbol [41,46],

$$
(a)_n = \frac{\Gamma(a+n)}{\Gamma(a)},
\tag{9}
$$

and $\Gamma(z)$ is the Euler's gamma function. The definite integral, which appears in Eq.(8), can be readily evaluated and we obtain,

$$
I \equiv \int_0^1 dt\; t^{c-1}(1-t)^{d-c-1}t^{mn} = \frac{\Gamma(d-c)\Gamma(c+mn)}{\Gamma(d+mn)}
$$
$$
= \frac{\Gamma(d-c)\Gamma(c)}{\Gamma(d)}\frac{\Gamma(c+mn)}{\Gamma(c)}\left(\frac{\Gamma(d+mn)}{\Gamma(d)}\right)^{-1}
$$
$$
= \frac{\Gamma(d-c)\Gamma(c)}{\Gamma(d)}\frac{(c)_{mn}}{(d)_{mn}}.
\tag{10}
$$

For integer values $m \in \mathbb{N}$ we can use the multiplication property of the Pochhammer symbol [46], i.e.,

$$
(a)_{k+mn} = (a)_k m^{mn}\prod_{j=0}^{m-1}\left(\frac{a+j+k}{m}\right)_n.
\tag{11}
$$

In our special case we have zero offset $k=0$, which simplifies the Pochhammer symbol into,

$$
(a)_{mn} = m^{mn}\prod_{j=0}^{m-1}\left(\frac{a+j}{m}\right)_n.
\tag{12}
$$

Thus, the integral given by Eq.(10) can be expressed in terms of the product of ratios of Pochhammer symbols,

$$
I = \frac{\Gamma(d-c)\Gamma(c)}{\Gamma(d)}\frac{\prod_{j=0}^{m-1}\left(\frac{c+j}{m}\right)_n}{\prod_{j=0}^{m-1}\left(\frac{d+j}{m}\right)_n}.
\tag{13}
$$

The form of the integral in Eq.(13) is particularly convenient for the evaluation of the sum in Eq.(8), which results in the generalized hypergeometric function of a higher order,

$$
\begin{aligned}
{}_A J_B &= \frac{\Gamma(d-c)\Gamma(c)}{\Gamma(d)} \sum_{n=0}^{\infty} \frac{(a_1)_n \cdots (a_A)_n}{(b_1)_n \cdots (b_B)_n} \frac{\prod_{j=0}^{m-1} \left(\frac{c+j}{m}\right)_n}{\prod_{j=0}^{m-1} \left(\frac{d+j}{m}\right)_n} \frac{z^n}{n!} \\
&= \frac{\Gamma(d-c)\Gamma(c)}{\Gamma(d)} {}_{A+m}F_{B+m} \left[ \begin{array}{cc} a_1,\ldots,a_A, & c_0,\cdots c_{m-1} \\ b_1,\ldots,b_B, & d_0,\cdots d_{m-1} \end{array} ; z \right],
\end{aligned}
\tag{14}
$$

where the constants $c_j$ and $d_j$ are given by $c_j = (c+j)/m$, and $d_j = (d+j)/m$ with index spanning $j \in [0, 1, \cdots, m-1]$. The comparison between Eq.(7) and Eq.(14) finishes the proof of the main result given by Eq.(6). Thus, we have shown that the generalized EIT of the hypergeometric function with power-law argument is a hypergeometric function of a higher order.

## 4 Fractional derivative of trigonometric functions

In sections 2 and 3 we have formulated the Caputo fractional derivative in terms of the generalized EIT and derived the main result for the generalized EIT of a hypergeometric function with a power-law argument. The goal of this section, as well as sections 5 to 8 is to apply the main result in Eq.(6) for the exact evaluation of the Caputo fractional derivative of a wide class of elementary functions. Specifically, in this section we obtain the Caputo fractional derivative of trigonometric and hyperbolic functions by means of the generalized EIT. We begin with the Caputo fractional derivative of $f(x) = \sin[(\beta x)^n]$ with integer power $n \in \mathbb{N}$,

$$
\begin{aligned}
&{}^C D_x^\alpha \left( \sin[(\beta x)^n] \right) \\
&= \frac{n \beta^n}{\Gamma(1-\alpha)} \int_0^x dt \, (x-t)^{-\alpha} t^{n-1} \cos[(\beta t)^n] \\
&= \frac{n \beta^n x^{n-\alpha}}{\Gamma(1-\alpha)} \int_0^1 dt \, (1-t)^{-\alpha} t^{n-1} \cos[(\beta x t)^n] \\
&= \frac{n \beta^n x^{n-\alpha}}{\Gamma(1-\alpha)} \int_0^1 dt \, (1-t)^{-\alpha} t^{n-1} {}_0F_1 \left[ ; \frac{1}{2}; -\frac{(\beta x t)^{2n}}{4} \right].
\end{aligned}
\tag{15}
$$

Unless otherwise stated, we set $\beta$ to be a constant parameter. In derivation of Eq.(15) we expressed cosine in terms of hypergeometric function Eq.(2),

$$
\cos[(\beta x)^n] = {}_0F_1 \left[ ; \frac{1}{2}; -\frac{(\beta x)^{2n}}{4} \right].
\tag{16}
$$

By comparing the general result Eq.(6) with the right hand side of Eq.(15) we obtain a system of linear equations for the variables $c, d, m$, and $z$ in terms of integer power $n$ and fractional parameter $\alpha$,

$$
\begin{aligned}
m &= 2n, \\
c - 1 &= n - 1, \\
d - c - 1 &= -\alpha, \\
z &= -\frac{(\beta x)^{2n}}{4},
\end{aligned}
\tag{17}
$$

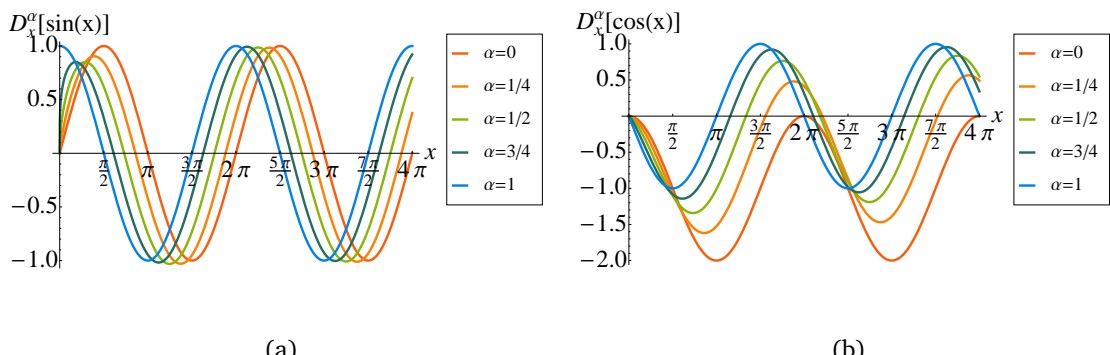

Figure 1: The Caputo fractional derivative of (a) $f(x) = \sin(x)$ and (b) $f(x) = \cos(x)$ for a range of orders of the fractional derivative, $0 \le \alpha \le 1$ that is shown in the legend. We shall note that the Caputo fractional derivative of the order $\alpha = 0$ is nothing but $^{C}D_{x}^{\alpha=0} = f(x) - f(0)$, so that $0^{\text{th}}$ order the Caputo fractional derivative always starts at the origin.

which can be readily solved, yielding $c = n$, $d = n + 1 - \alpha$. Thus, we obtain the Caputo fractional derivative of sine,

$$
\begin{aligned}
^{C}D_{x}^{\alpha}\left(\sin[(\beta x)^{n}]\right) &= \beta^{n}x^{n-\alpha}\frac{\Gamma(n+1)}{\Gamma(n+1-\alpha)} \\
&\times {}_{2n}F_{2n+1}\left[\begin{array}{c} n/2n, (n+1)/2n, \cdots (3n-1)/2n \\ 1/2, (n+1-\alpha)/2n, \cdots (3n-\alpha)/2n \end{array}; -\frac{(\beta x)^{2n}}{4}\right] \\
&= \beta^{n}x^{n-\alpha}\frac{\Gamma(n+1)}{\Gamma(n+1-\alpha)}{}_{2n-1}F_{2n}\left[\begin{array}{c} (n+1)/2n, \cdots (3n-1)/2n \\ (n+1-\alpha)/2n, \cdots (3n-\alpha)/2n \end{array}; -\frac{(\beta x)^{2n}}{4}\right].
\end{aligned}
\tag{18}
$$

In the special case of $n = 1$ the general form of the Caputo fractional derivative can be simplified to [2,7],

$$
^{C}D_{x}^{\alpha}[\sin(\beta x)] = \frac{\beta x^{1-\alpha}}{\Gamma(2-\alpha)}{}_{1}F_{2}\left[\begin{array}{c} 1 \\ (2-\alpha)/2, (3-\alpha)/2 \end{array}; -\frac{\beta^{2}x^{2}}{4}\right].
\tag{19}
$$

Next, we turn to the Caputo fractional derivative of $f(x) = \cos[(\beta x)^{n}]$, and bring it into the generalized EIT form,

$$
\begin{aligned}
&^{C}D_{x}^{\alpha}\left(\cos[(\beta x)^{n}]\right) \\
&= -\frac{n\beta^{n}}{\Gamma(1-\alpha)}\int_{0}^{x}dt\,(x-t)^{-\alpha}t^{n-1}\sin[(\beta t)^{n}] \\
&= -\frac{n\beta^{n}x^{n-\alpha}}{\Gamma(1-\alpha)}\int_{0}^{1}dt\,(1-t)^{-\alpha}t^{n-1}\sin[(\beta xt)^{n}] \\
&= -\frac{n\beta^{2n}x^{2n-\alpha}}{\Gamma(1-\alpha)}\int_{0}^{1}dt\,(1-t)^{-\alpha}t^{2n-1}{}_{0}F_{1}\left[;\frac{3}{2}; -\frac{(\beta xt)^{2n}}{4}\right],
\end{aligned}
\tag{20}
$$

where we have used the well-known relation between sine and the hypergeometric function [41] (see Eq.(2)),

$$
\sin[(\beta x)^{n}] = (\beta x)^{n}\,{}_{0}F_{1}\left(;\frac{3}{2}; -\frac{(\beta x)^{2n}}{4}\right).
\tag{21}
$$

Direct comparison of the right hand side of Eq.(20) with the generalized Euler's transform results in the system of linear equations,

$$
\begin{aligned}
m &= 2n, \\
c - 1 &= 2n - 1, \\
d - c - 1 &= -\alpha, \\
z &= -\frac{(\beta x)^{2n}}{4}.
\end{aligned}
\tag{22}
$$

We immediately obtain $c = 2n$ and $d = 2n + 1 - \alpha$, and thus the Caputo fractional derivative of cosine is given by the generalized hypergeometric function,

$$
\begin{aligned}
{}^{C}D_x^\alpha \left( \cos[(\beta x)^n] \right) &= -\beta^{2n} x^{2n-\alpha} \frac{n\Gamma(2n)}{\Gamma(2n+1-\alpha)} \\
&\times {}_{2n}F_{2n+1} \left[ \begin{array}{c} 2n/2n, \cdots (4n-1)/2n \\ 3/2, (2n+1-\alpha)/2n, \cdots (4n-\alpha)/2n \end{array} ; -\frac{(\beta x)^{2n}}{4} \right] \\
&= -\frac{\beta^{2n} x^{2n-\alpha}}{2} \frac{\Gamma(2n+1)}{\Gamma(2n+1-\alpha)} \, {}_{2n}F_{2n+1} \left[ \begin{array}{c} 1, (2n+1)/2n, \cdots (4n-1)/2n \\ 3/2, (2n+1-\alpha)/2n, \cdots (4n-\alpha)/2n \end{array} ; -\frac{(\beta x)^{2n}}{4} \right].
\end{aligned}
\tag{23}
$$

In the special case of $n = 1$, the Caputo fractional derivative of cosine can be significantly simplified [7],

$$
{}^{C}D_x^\alpha [\cos(\beta x)] = -\frac{\beta^2 x^{2-\alpha}}{\Gamma(3-\alpha)} \, {}_1F_2 \left[ \begin{array}{c} 1 \\ (3-\alpha)/2, (4-\alpha)/2 \end{array} ; -\frac{\beta^2 x^2}{4} \right].
\tag{24}
$$

By combing results in Eq.(19) and Eq. (24), we obtain the Caputo fractional derivative of a plane wave,

$$
\begin{aligned}
{}^{C}D_x^\alpha &[\exp(i\beta x)] \\
&= -\frac{\beta^2 x^{2-\alpha}}{\Gamma(3-\alpha)} \, {}_1F_2 \left[ \begin{array}{c} 1 \\ (3-\alpha)/2, (4-\alpha)/2 \end{array} ; -\frac{\beta^2 x^2}{4} \right] \\
&+ i \frac{\beta x^{1-\alpha}}{\Gamma(2-\alpha)} \, {}_1F_2 \left[ \begin{array}{c} 1 \\ (2-\alpha)/2, (3-\alpha)/2 \end{array} ; -\frac{\beta^2 x^2}{4} \right].
\end{aligned}
\tag{25}
$$

In order to obtain the Caputo fractional derivative of hyperbolic functions, we employ the imaginary arguments, i.e., $\sin[i(\beta x)] = i \sinh(\beta x)$ and $\cos[i(\beta x)] = \cosh(\beta x)$. Thus, we immediately obtain,

$$
\begin{aligned}
{}^{C}D_x^\alpha \left( \sinh[(\beta x)^n] \right) &= \beta^n x^{n-\alpha} \frac{\Gamma(n+1)}{\Gamma(n+1-\alpha)} \\
&\times {}_{2n-1}F_{2n} \left[ \begin{array}{c} (n+1)/2n, (n+2)/2n, \cdots (3n-1)/2n \\ (n+1-\alpha)/2n, \cdots (3n-\alpha)/2n \end{array} ; \frac{(\beta x)^{2n}}{4} \right],
\end{aligned}
\tag{26}
$$

and

$$
\begin{aligned}
{}^{C}D_x^\alpha \left( \cosh[(\beta x)^n] \right) &= \frac{\beta^{2n} x^{2n-\alpha}}{2} \frac{\Gamma(2n+1)}{\Gamma(2n+1-\alpha)} \\
&\times {}_{2n}F_{2n+1} \left[ \begin{array}{c} 1, (2n+1)/2n, \cdots (4n-1)/2n \\ 3/2, (2n+1-\alpha)/2n, \cdots (4n-\alpha)/2n \end{array} ; \frac{(\beta x)^{2n}}{4} \right].
\end{aligned}
\tag{27}
$$

Thus, the Caputo fractional derivative of harmonic functions is given by the generalized hypergeometric function, which, in the most general case, cannot be reduced to elementary functions.

## 5 The Caputo fractional derivative of inverse trigonometric functions

In this section we will apply the main result in Eq.(6) for the exact evaluation of the Caputo fractional derivative of inverse trigonometric functions. We begin with the Caputo fractional derivative of $f(x) = \arcsin[(\beta x)^n]$ with integer power $n \in \mathbb{N}$,

$$
{}^C D_x^\alpha (\arcsin[(\beta x)^n]) \tag{28}
$$
$$
= \frac{n\beta^n}{\Gamma(1-\alpha)} \int_0^x dt \, (x-t)^{-\alpha} \frac{t^{n-1}}{\sqrt{1-(\beta x)^{2n}}}
$$
$$
= \frac{n\beta^n x^{n-\alpha}}{\Gamma(1-\alpha)} \int_0^1 dt \, (1-t)^{-\alpha} \frac{t^{n-1}}{\sqrt{1-(\beta x t)^{2n}}}
$$
$$
= \frac{n\beta^n x^{n-\alpha}}{\Gamma(1-\alpha)} \int_0^1 dt \, (1-t)^{-\alpha} t^{n-1} {}_2F_1\left[1,\frac{1}{2};1;(\beta x t)^{2n}\right],
$$

where we have used the well-known relation [41],

$$
(1+\xi)^k = {}_2F_1[-k,1;1;-\xi], \tag{29}
$$

with $k = -1/2$ and $\xi = -(\beta x t)^{2n}$. Direct comparison of the right hand side of Eq.(28) with the general result given by Eq.(6) leads to the system of linear equations,

$$
\begin{aligned}
m &= 2n, \\
c-1 &= n-1, \\
d-c-1 &= -\alpha, \\
z &= (\beta x)^{2n}.
\end{aligned} \tag{30}
$$

With coefficients $c = n$ and $d = n+1-\alpha$ we immediately obtain,

$$
{}^C D_x^\alpha (\arcsin[(\beta x)^n]) = \beta^n x^{n-\alpha} \frac{\Gamma(n+1)}{\Gamma(n+1-\alpha)} \tag{31}
$$
$$
\times {}_{2n+2}F_{2n+1}\left[\begin{array}{c} 1,1/2,n/2n,\cdots(3n-1)/2n \\ 1,(n+1-\alpha)/2n,\cdots(3n-\alpha)/2n \end{array} ;(\beta x)^{2n}\right]
$$
$$
= \beta^n x^{n-\alpha} \frac{\Gamma(n+1)}{\Gamma(n+1-\alpha)} {}_{2n+1}F_{2n}\left[\begin{array}{c} 1/2,n/2n,(n+2)/2n,\cdots(3n-1)/2n \\ (n+1-\alpha)/2n,\cdots(3n-\alpha)/2n \end{array} ;(\beta x)^{2n}\right].
$$

From the definition of the Caputo fractional derivative given by Eq.(1), we immediately notice that its value for $f(x) = \arccos[(\beta x)^n]$ is opposite in sign to the Caputo fractional derivative of $f(x) = \arcsin[(\beta x)^n]$,

$$
{}^C D_x^\alpha (\arccos[(\beta x)^n]) = -{}^C D_x^\alpha (\arcsin[(\beta x)^n]). \tag{32}
$$

In the special case of $n = 1$ we obtain,

$$
{}^C D_x^\alpha (\arcsin[\beta x]) = \beta x^{1-\alpha} \frac{1}{\Gamma(2-\alpha)} {}_3F_2\left[\begin{array}{c} 1/2,1/2,1 \\ (2-\alpha)/2,(3-\alpha)/2 \end{array} ;(\beta x)^2\right]. \tag{33}
$$

The analogous calculation leads to the Caputo fractional derivative of $f(x) = \arctan[(\beta x)^n]$,

$$
{}^C D_x^\alpha (\arctan[(\beta x)^n]) = \beta^n x^{n-\alpha} \frac{\Gamma(n+1)}{\Gamma(n+1-\alpha)} \tag{34}
$$
$$
\times {}_{2n+1}F_{2n}\left[\begin{array}{c} 1,1/2,(n+1)/2n,\cdots(3n-1)/2n \\ (n+1-\alpha)/2n,\cdots(3n-\alpha)/2n \end{array} ;-(\beta x)^{2n}\right].
$$

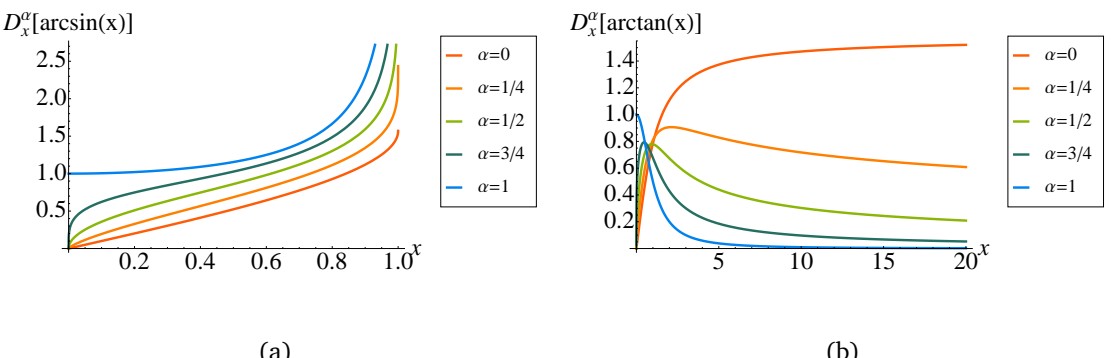

(a)                                                              (b)

Figure 2: The Caputo fractional derivative of (a) $f(x) = \arcsin(x)$ and (b) $f(x) = \arctan(x)$ for a range of orders of the fractional derivative, $0 \leq \alpha \leq 1$, shown in the legend. The Caputo fractional derivative of $f(x) = \arccos(x)$ and $f(x) = \text{arccot}(x)$ differ by a negative sign from the Caputo fractional derivative of $f(x) = \arcsin(x)$ and $f(x) = \arctan(x)$, correspondingly.

This immediately leads to the Caputo fractional derivative of $f(x) = \text{arccot}[(\beta x)^n]$,

$${}^{C}D_x^{\alpha}\left(\text{arccot}[(\beta x)^n]\right) = -\,{}^{C}D_x^{\alpha}\left(\arctan[(\beta x)^n]\right). \tag{35}$$

In the special case of $n = 1$ we obtain,

$${}^{C}D_x^{\alpha}\left(\arctan[\beta x]\right) = \beta x^{1-\alpha}\frac{1}{\Gamma(2-\alpha)}{}_3F_2\left[\begin{array}{c}1/2,1,1\\(2-\alpha)/2,(3-\alpha)/2\end{array};-(\beta x)^2\right]. \tag{36}$$

We shall note that we were able to obtain exact analytical results for the Caputo fractional derivative of $f(x) = \arcsin[(\beta x)]$ and $f(x) = \arctan[(\beta x)]$ due to the fact that their derivatives are expressed in terms of the generalized hypergeometric function with a *power-law* argument. Unfortunately, the general result in Eq.(6) cannot be directly applied to $f(x) = \tan[(\beta x)]$, since its representation in terms of hypergeometric functions involves the corresponding ratio of Eq.(21) and Eq.(16), which precludes us from the exact evaluation by means of the formula in Eq.(6). Thus, it naturally establishes the limit of the applicability of the generalized EIT given by Eq.(6) for the exact evaluation of the Caputo fractional derivative.

## 6 The Caputo fractional derivative of the Gaussian function

In this section our goal is to obtain the exact result for the Caputo fractional derivative of the Gaussian function and, in the most general case, an exponential with a power-law argument. We begin with the Caputo fractional derivative of a power-law exponential, $f(x) = \exp[-(\beta x)^n]$, with integer power $n \in \mathbb{N}$,

$$\begin{aligned}
&{}^{C}D_x^{\alpha}\left(\exp[-(\beta x)^n]\right)\\
&= -\frac{n\beta^n}{\Gamma(1-\alpha)}\int_0^x dt\,(x-t)^{-\alpha}t^{n-1}\exp[-(\beta t)^n]\\
&= -\frac{n\beta^n x^{n-\alpha}}{\Gamma(1-\alpha)}\int_0^1 dt\,(1-t)^{-\alpha}t^{n-1}\exp[-(\beta x t)^n]\\
&= -\frac{n\beta^n x^{n-\alpha}}{\Gamma(1-\alpha)}\int_0^1 dt\,(1-t)^{-\alpha}t^{n-1}{}_1F_1[1;1;-(\beta x t)^n],
\end{aligned} \tag{37}$$

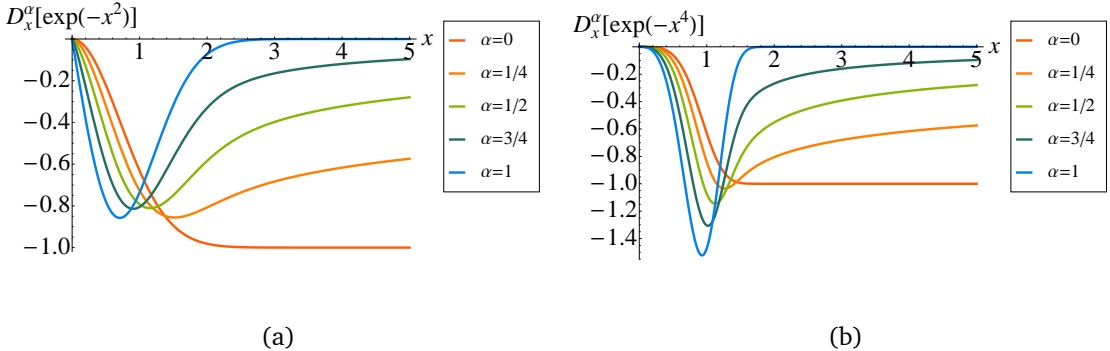

Figure 3: The Caputo fractional derivative of (a) $f(x) = \exp(-x^2)$ and (b) $f(x) = \exp(-x^4)$ for a range of orders of the fractional derivative, $0 \leq \alpha \leq 1$, shown in the legend. We shall note that the Caputo fractional derivative of the order $\alpha = 0$ is nothing but $^{C}D_x^{\alpha=0} = f(x) - f(0)$ so that $0^{\text{th}}$ order of the Caputo fractional derivative is shifted by its value at the origin.

where we have expressed the exponential in terms of the hypergeometric function according to [41],

$$\exp[-(\beta x)^n] = {}_1F_1[1;1;-(\beta x)^n]. \tag{38}$$

In this case, the system of linear equations that reduces the general result in Eq.(6) to the right hand side of Eq.(37) is given by

$$
\begin{aligned}
m &= n, \\
c - 1 &= n - 1, \\
d - c - 1 &= -\alpha, \\
z &= -(\beta x)^n,
\end{aligned}
\tag{39}
$$

which leads to $c = n$ and $d = n + 1 - \alpha$. Thus, we obtain the exact result for the Caputo fractional derivative of an exponential with a power-law argument,

$$
{}^{C}D_x^{\alpha}(\exp[-(\beta x)^n]) = -\beta^n x^{n-\alpha} \frac{\Gamma(n+1)}{\Gamma(n+1-\alpha)}
$$
$$
\times {}_nF_n\left[\begin{matrix} 1, (n+1)/n, \cdots (2n-1)/n \\ (n+1-\alpha)/n, (n+2-\alpha)/n, \cdots (2n-\alpha)/n \end{matrix}; -(\beta x)^n\right].
\tag{40}
$$

In the special case of $n = 2$ we obtain the Caputo fractional derivative of the Gaussian function,

$$
{}^{C}D_x^{\alpha}[\exp[-(\beta x)^2]] = \frac{-2\beta^2 x^{2-\alpha}}{\Gamma(3-\alpha)} {}_2F_2\left[\begin{matrix} 1, 3/2 \\ (3-\alpha)/2, (4-\alpha)/2 \end{matrix}; -(\beta x)^2\right].
\tag{41}
$$

## 7 The Caputo fractional derivative of the Lorentzian function

The goal of this section is to evaluate the Caputo fractional derivative of the Lorentzian function, which plays an important role in quantum optics [47], atomic spectroscopy [48] and quantum electrodynamics [49]. The Lorentzian function is defined as

$$
f_L(x, \gamma) = \frac{1}{\pi} \frac{\gamma/2}{(x^2 + \gamma^2/4)}.
\tag{42}
$$

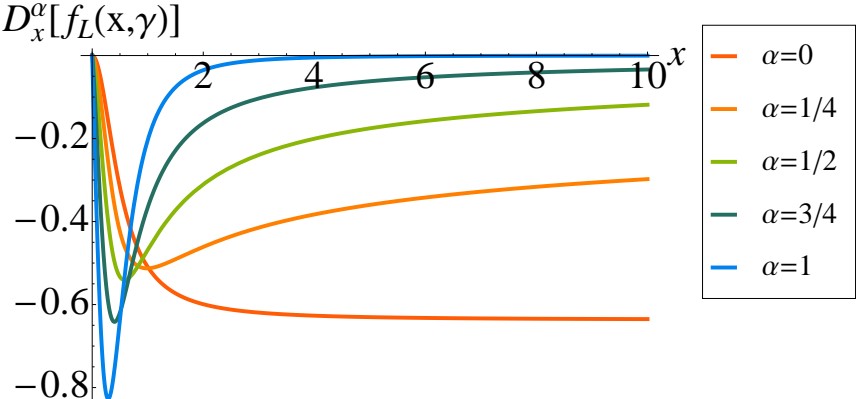

Figure 4: The Caputo fractional derivative of the Lorentzian function $f_L(x,\gamma)$, defined in Eq.(42), for a range of orders of the fractional derivative, $0 \le \alpha \le 1$, shown in the legend. We shall note that the Caputo fractional derivative of the order $\alpha = 0$ is nothing but ${}^C D_x^{\alpha=0} = f(x) - f(0)$ so that $0^{\text{th}}$ order fractional derivative is shifted by its value at the origin.

Thus, the Caputo fractional derivative of the Lorentzian function becomes,

$$
{}^C D_x^\alpha (f_L(x,\gamma)) \tag{43}
$$
$$
= -\frac{\gamma}{\pi \Gamma(1-\alpha)} \int_0^x dt \, (x-t)^{-\alpha} \frac{t}{\left(t^2 + \frac{\gamma^2}{4}\right)^2}
$$
$$
= -\frac{x^{2-\alpha}\gamma}{\pi \Gamma(1-\alpha)} \int_0^1 dt \, t(1-t)^{-\alpha} \frac{1}{\left((xt)^2 + \frac{\gamma^2}{4}\right)^2}
$$
$$
= -\frac{16}{\gamma^4} \frac{x^{2-\alpha}\gamma}{\pi \Gamma(1-\alpha)} \int_0^1 dt \, t(1-t)^{-\alpha} {}_2F_1\left[2,1;1;-4(xt/\gamma)^2\right],
$$

where we have employed Eq.(29) with $k = -2$ and $\xi = 4(xt/\gamma)^2$ to represent the first derivative of the Lorentzian function in terms of the hypergeometric function. In this particular case, the system of linear equations that reduces the general result in Eq.(6) to the right hand side of Eq.(43) is given by

$$
\begin{aligned}
m &= 2, \\
c - 1 &= 1, \\
d - c - 1 &= -\alpha, \\
z &= -4(x/\gamma)^2, \tag{44}
\end{aligned}
$$

which leads to $c = 2$ and $d = 3 - \alpha$. Thus, we obtain the exact result for the Caputo fractional derivative of the Lorentzian function,

$$
{}^C D_x^\alpha (f_L(x,\gamma)) = -\frac{16 x^{2-\alpha}}{\pi \gamma^3 \Gamma(3-\alpha)} {}_3F_2\left[\begin{array}{c} 1, 3/2, 2 \\ (3-\alpha)/2, (4-\alpha)/2 \end{array} ; -4\left(\frac{x}{\gamma}\right)^2\right]. \tag{45}
$$

## 8 The Caputo fractional derivative of a shifted polynomial

In this section we will apply the main result in Eq.(6) for the exact evaluation of the Caputo fractional derivative of a shifted polynomial. This result is important for the exact evaluation

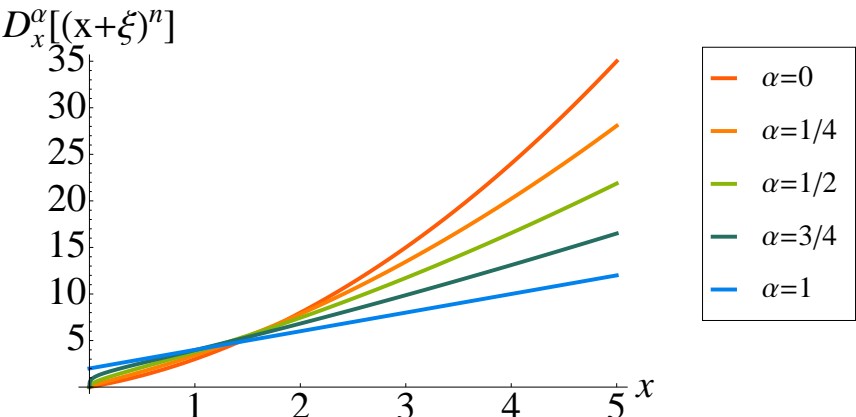

Figure 5: The Caputo fractional derivative of shifted polynomial $f(x) = (x + \xi)^n$ for a range of orders of the fractional derivative, $0 \le \alpha \le 1$, shown in the legend. For concreteness we have chosen $\xi = 1$ and $n = 2$.

of the Caputo fractional derivative of a number of functions that cannot be expressed in terms of a single generalized hypergeometric function with a polynomial argument. Despite this limitation, one can still expand these functions in the Taylor series and apply the generalized EIT to each term in the expansion. In order to apply this method, we shall find the Caputo fractional derivative of a shifted polynomial function, i.e., $f(x) = (x+\xi)^n$, with integer power $n \in \mathbb{N}$,

$$
\begin{aligned}
{}^C&D_x^\alpha((x+\xi)^n) \\
&= \frac{n}{\Gamma(1-\alpha)} \int_0^x dt\, (x-t)^{-\alpha}(t+\xi)^{n-1} \\
&= \frac{\xi^{n-1}}{x^{\alpha-1}} \frac{n}{\Gamma(1-\alpha)} \int_0^1 dt\, (1-t)^{-\alpha}\left(1+\frac{x}{\xi}t\right)^{n-1} \\
&= \frac{\xi^{n-1}}{x^{\alpha-1}} \frac{n}{\Gamma(1-\alpha)} \int_0^1 dt\, (1-t)^{-\alpha}{}_2F_1[1-n,1;1;-xt/\xi].
\end{aligned}
\tag{46}
$$

In this case, the system of linear equations that reduces the general result in Eq.(6) to the right hand side of Eq.(46) is given by

$$
\begin{aligned}
m &= 1, \\
c - 1 &= 0, \\
d - c - 1 &= -\alpha, \\
z &= -x/\xi,
\end{aligned}
\tag{47}
$$

which leads to $c = 1$ and $d = 2 - \alpha$. Thus, we obtain the exact result for the Caputo fractional derivative of a shifted polynomial,

$$
\begin{aligned}
{}^C D_x^\alpha[(x+\xi)^n] &= \frac{\xi^{n-1}}{x^{\alpha-1}} \frac{n}{\Gamma(2-\alpha)} {}_3F_2\left[\begin{array}{c} 1,1,1-n \\ 1,(2-\alpha) \end{array}; -\frac{x}{\xi}\right] \\
&= \frac{\xi^{n-1}}{x^{\alpha-1}} \frac{n}{\Gamma(2-\alpha)} {}_2F_1\left[\begin{array}{c} 1,1-n \\ (2-\alpha) \end{array}; -\frac{x}{\xi}\right].
\end{aligned}
\tag{48}
$$

## 9 Equivalence between the Liouville–Caputo and Fourier fractional derivatives

In this section we will consider the Liouville–Caputo fractional derivative of the fractional order $0 \le \alpha \le 1$ which can be obtained from the Caputo fractional derivative, which is defined in Eq.(1), by extending the lower integration limit from zero to negative infinity [7],

$$^{\mathrm{LC}}D_x^\alpha f(x) = \frac{1}{\Gamma(1-\alpha)} \int_{-\infty}^x dt \, (x-t)^{-\alpha} \frac{df(t)}{dt}. \tag{49}$$

Our goal is to establish the connection between the Caputo and Liouville–Caputo fractional derivatives. But first, we show that the Liouville–Caputo fractional derivative is completely equivalent to the Fourier fractional derivative, defined as [7]

$$^{\mathrm{F}}D_x^\alpha f(x) = \frac{1}{2\pi} \int_{-\infty}^\infty dk \, \widehat{f}(k)(-ik)^\alpha \exp(-ikx), \tag{50}$$

where $\widehat{f}(k)$ is the Fourier image of the function $f(x)$, i.e.,

$$f(t) = \frac{1}{2\pi} \int_{-\infty}^\infty dk \, \widehat{f}(k) \exp(-ikt). \tag{51}$$

In order to prove the equivalence between the Liouville–Caputo and Fourier fractional derivatives, we substitute the Fourier image given by Eq.(51) into Eq.(49), which results in,

$$^{\mathrm{LC}}D_x^\alpha f(x) = \frac{1}{2\pi} \frac{1}{\Gamma(1-\alpha)} \int_{-\infty}^\infty dk \, (-ik)\widehat{f}(k) \int_{-\infty}^x dt \, (x-t)^{-\alpha} \exp(-ikt). \tag{52}$$

By shifting the variable $t \to x - t$, the inner integral in Eq.(52) becomes,

$$I = \int_{-\infty}^x dt \, (x-t)^{-\alpha} \exp(-ikt) = \exp(-ikx) \int_0^\infty dt \, t^{-\alpha} \exp(ikt). \tag{53}$$

Now we perform a change of variable, $t = i\eta/k$, which allows us to evaluate the integral in Eq.(53) in terms of the Euler gamma function,

$$\begin{aligned} I &= \exp(-ikx)(-ik)^{\alpha-1} \int_0^{-i\infty} d\eta \, \eta^{-\alpha} \exp(-\eta) \\ &= \exp(-ikx)(-ik)^{\alpha-1} \int_0^\infty d\eta \, \eta^{-\alpha} \exp(-\eta) \\ &= \exp(-ikx)(-ik)^{\alpha-1} \Gamma(1-\alpha), \end{aligned} \tag{54}$$

where we have used the Cauchy residue theorem and the definition of the Euler gamma function. Combing Eq.(52) with Eq.(54) we finally prove the equivalence between the Liouville–Caputo and Fourier fractional derivatives,

$$^{\mathrm{LC}}D_x^\alpha f(x) = \frac{1}{2\pi} \frac{1}{\Gamma(1-\alpha)} \tag{55}$$

$$\times \int_{-\infty}^\infty dk \, (-ik)\widehat{f}(k) \exp(-ikx)(-ik)^{\alpha-1}\Gamma(1-\alpha)$$

$$= \frac{1}{2\pi} \int_{-\infty}^\infty dk \, \widehat{f}(k)(-ik)^\alpha \exp(-ikx) = {}^{\mathrm{F}}D_x^\alpha f(x). \tag{56}$$

Hence, the Liouville–Caputo and Fourier definitions given by Eq.(49) and Eq.(50), correspondingly, are alternative, but, nevertheless, completely equivalent forms of a fractional derivative.

## 10 Correspondence between the Caputo and Liouville–Caputo fractional derivative

In this section we show the correspondence between the Caputo and Liouville–Caputo fractional derivatives of elementary functions in the infinite limit of their arguments. First, the equivalence between Liouville–Caputo and Fourier fractional derivatives formulated by Eq.(55) allows us to readily evaluate their values for the harmonic functions, e.g, [1,7]

$$
\begin{aligned}
{}^{\text{LC}}D_t^\alpha[\sin(\beta t)] &= \beta^\alpha \sin\left(\beta t + \frac{\pi\alpha}{2}\right), \\
{}^{\text{LC}}D_t^\alpha[\exp(i\beta t)] &= \beta^\alpha \exp\left(i\beta t + \frac{i\pi\alpha}{2}\right).
\end{aligned}
\tag{57}
$$

Thus, the Liouville–Caputo and Fourier fractional derivatives of the order $\alpha$ of harmonic functions, aside from the factor $\beta^\alpha$, effectively introduce a shift in the argument's phase given by $\pi\alpha/2$. In the previous sections 4 to 8 we have proven that the Caputo fractional derivative of elementary functions is expressed in terms of generalized hypergeometric functions, which, in the most general case, cannot be simplified to elementary functions. However, in the infinite limit of the argument of the hypergeometric functions, we can expand it in a Taylor series, which results in

$$
\begin{aligned}
&{}^{\text{C}}D_x^\alpha[\sin(\beta t)] \\
&= \beta^\alpha \sin\left(\beta t + \frac{\pi\alpha}{2}\right) + t^{-\alpha}\left(\frac{1}{\beta t\Gamma(-\alpha)} + \mathcal{O}\left(\frac{1}{t^3}\right)\right)\bigg|_{t\to\infty} \\
&= {}^{\text{LC}}D_x^\alpha[\sin(\beta t)].
\end{aligned}
\tag{58}
$$

Thus, we have shown that in the infinite limit of the argument of elementary functions all three definitions of a fractional derivative - Caputo, Liouville–Caputo, and Fourier - converge to the same result given by elementary functions.

The final goal of this section is to derive the Liouville–Caputo, or equivalently the Fourier fractional derivative, of the Gaussian function. First, we shall point out that the Fourier fractional derivative of the Gaussian function was derived previously [7], and was given in terms of the Kummer hypergeometric functions,

$$
\begin{aligned}
&{}^{\text{LC}}D_x^\alpha[\exp(-\beta x^2)] \\
&= \frac{1}{2\pi}\sqrt{\frac{\pi}{\beta}}\int_{-\infty}^{\infty} dk\,(-ik)^\alpha \exp\left(-\frac{k^2}{4\beta}\right)\exp(-ikx) \\
&= \frac{2^\alpha \beta^{\alpha/2}}{\sqrt{\pi}}\left\{\cos\left(\frac{\pi\alpha}{2}\right)\Gamma\left(\frac{\alpha+1}{2}\right){}_1F_1\left(\frac{\alpha+1}{2};\frac{1}{2};-x^2\beta\right)\right. \\
&\left. -x\alpha\sqrt{\beta}\sin\left(\frac{\pi\alpha}{2}\right)\Gamma\left(\frac{\alpha}{2}\right){}_1F_1\left(\frac{\alpha}{2}+1;\frac{3}{2};-x^2\beta\right)\right\}.
\end{aligned}
\tag{59}
$$

Our goal is to prove that the Liouville–Caputo fractional derivative of the fractional order $\alpha$ of the Gaussian function is given by a single Hermite polynomial with a fractional index $\alpha$, i.e.,

$$
{}^{\text{LC}}D_x^\alpha[\exp(-\beta x^2)] = \beta^{\alpha/2}\exp\left(-\beta x^2\right)H_\alpha\left(-\sqrt{\beta}x\right).
\tag{60}
$$

Below we show the equivalence between Eq.(59) and Eq.(60). The Hermite polynomials, $H_\alpha(x)$, of a positive non-integer order $\alpha$ are directly related to the parabolic cylinder function, $D_\alpha(z)$. This correspondence can be shown via the integral representation of the Hermite polynomials, [41, 42, 50, 51],

$$
H_\alpha(x) = \frac{2^{\alpha+1}}{\sqrt{\pi}}e^{x^2}\int_0^\infty e^{-t^2}t^\alpha\cos\left(2xt - \frac{\pi\alpha}{2}\right)dt = \exp\left(\frac{x^2}{2}\right)2^{\alpha/2}D_\alpha\left(\sqrt{2}x\right).
\tag{61}
$$

On the order hand, the parabolic cylinder function can be expressed in terms of the Tricomi confluent hypergeometric function [41, 42, 50, 51]

$$D_\alpha(x) = \exp\left(-\frac{x^2}{4}\right) 2^{\alpha/2} U\left(-\frac{\alpha}{2}, \frac{1}{2}, \frac{x^2}{2}\right). \tag{62}$$

Thus, the Hermite polynomials of non-integer order $\alpha$ are directly related to the Tricomi confluent hypergeometric function,

$$H_\alpha(x) = 2^\alpha U\left(-\frac{\alpha}{2}, \frac{1}{2}, x^2\right). \tag{63}$$

Further, the Tricomi confluent hypergeometric function can be expressed in terms of the Kummer confluent hypergeometric functions as [42]

$$U(a, b, z) = \frac{\Gamma(1-b)}{\Gamma(a+1-b)}{}_1F_1(a, b, z) + \frac{\Gamma(b-1)}{\Gamma(a)} z^{1-b} {}_1F_1(a+1-b, 2-b, z). \tag{64}$$

In our case we have, $a = -\frac{\alpha}{2}$, $b = \frac{1}{2}$, and $z = x^2$, which results in

$$U\left(-\frac{\alpha}{2}; \frac{1}{2}; x^2\right) = \frac{\Gamma\left(\frac{1}{2}\right)}{\Gamma\left(\frac{1-\alpha}{2}\right)}{}_1F_1\left[-\frac{\alpha}{2}; \frac{1}{2}; x^2\right] + \frac{\Gamma\left(-\frac{1}{2}\right)}{\Gamma\left(\frac{-\alpha}{2}\right)} x \, {}_1F_1\left[\frac{1-\alpha}{2}; \frac{3}{2}; x^2\right]. \tag{65}$$

The well-known Euler's reflection formula [41],

$$\Gamma(1-z)\Gamma(z) = \frac{\pi}{\sin(\pi z)}, \tag{66}$$

along with the Kummer's relation for the hypergeometric function [41, 42],

$${}_1F_1(a; b; x^2) = e^{x^2} {}_1F_1(b-a; b; -x^2) \tag{67}$$

immediately lead to,

$$\begin{aligned} 2^{-\alpha} H_\alpha(x) &= U\left(-\frac{\alpha}{2}; \frac{1}{2}; x^2\right) \\ &= e^{x^2} \frac{1}{\sqrt{\pi}} \left(\cos(\pi\alpha/2)\Gamma\left(\frac{1+\alpha}{2}\right) {}_1F_1\left[\frac{1+\alpha}{2}; \frac{1}{2}; -x^2\right]\right. \\ &\quad \left. + \alpha x \sin(\pi\alpha/2)\Gamma\left(\frac{\alpha}{2}\right) {}_1F_1\left[1+\frac{\alpha}{2}; \frac{3}{2}; -x^2\right]\right). \end{aligned} \tag{68}$$

Lastly, by re-scaling the argument, $x \to -\sqrt{\beta} x$, we obtain the final result,

$$\begin{aligned} e^{-\beta x^2} &H_\alpha(-\sqrt{\beta} x) \\ &= \frac{2^\alpha}{\sqrt{\pi}} \left\{\cos\left(\frac{\pi\alpha}{2}\right)\Gamma\left(\frac{\alpha+1}{2}\right) {}_1F_1\left(\frac{\alpha+1}{2}; \frac{1}{2}; -x^2\beta\right)\right. \\ &\quad \left. - x\alpha\sqrt{\beta} \sin\left(\frac{\pi\alpha}{2}\right)\Gamma\left(\frac{\alpha}{2}\right) {}_1F_1\left(\frac{\alpha}{2}+1; \frac{3}{2}; -x^2\beta\right)\right\}, \end{aligned} \tag{69}$$

which proves the equivalence between Eq.(59) and Eq.(60). Thus, the Liouville–Caputo, or equivalently, the Fourier fractional derivative of the Gaussian function is nothing but a single Hermite polynomial of a fractional index $\alpha$, which, in turn, is the order of the fractional derivative.

## 11 Conclusions

In this paper, we considered the Caputo fractional derivative and found its exact analytical values for a broad class of elementary functions. These results were made possible by representing the Caputo fractional derivative in terms of the generalized Euler's integral transform (EIT). This transform formulates a definite integral of the beta-type distribution, combined with the hypergeometric function with a polynomial argument, in terms of a single hypergeometric function of a higher order. We presented a proof of the generalized EIT and directly applied it to the exact evaluation of the Caputo fractional derivative of an extensive class of functions, provided that they can be expressed in terms of a generalized hypergeometric function with a power-law argument. The generalized EIT effectively reduces the evaluation of the Caputo fractional derivative to a system of linear equation which can be readily solved. We found that the Caputo fractional derivative of elementary functions is given by the generalized hypergeometric function. Furthermore, we established that the obtained result for the Caputo fractional derivative cannot be reduced to elementary functions in contrast to both Liouville–Caputo and Fourier fractional derivatives. However, we found that in the infinite limit of the argument of elementary functions, the Caputo, Liouville–Caputo, and Fourier fractional derivatives - converge to the same analytical result given by elementary functions. Finally, we demonstrated the complete equivalence between the Liouville–Caputo and Fourier fractional derivative that define fractional derivative in the configuration and momentum space, correspondingly.

## Acknowledgments

The authors would like to thank Daniel Jaschke and Marc Valdez for numerous and fruitful discussions. The authors acknowledge support from the US National Science Foundation under grant numbers PHY-1520915, OAC-1740130, DMR- 1407962, and the US Air Force Office of Scientific Research grant number FA9550- 14-1-0287. This work was performed in part at the Aspen Center for Physics, which is supported by the US National Science Foundation grant PHY-1607611.

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
