# Peer review of "Exact results for a fractional derivative of elementary functions"

_SciPost Physics, doi:SciPost Phys. 4, 029 (2018)_

## Round 2 · Referee Report · Anonymous (Referee 1) · 2018-2-15

Strengths

An interesting connection between fractional derivatives and special functions, specifically some algebraic functions.

Weaknesses

There are some misprints that must be corrected.

Report

See below: requested changes

Requested changes

The definition given by Eq.(1) is valid for $0 < \alpha < 1$ with $\alpha \in \mathbb{R}$. The cases $\alpha =0$ and $\alpha = 1$ must be treat as separated cases. See Kilbas, Srivastava and Trujillo, reference [1], page 92. This occours also on pages 4, 7, 10, 11, 12, 13, specifically to obtain the graphycs.

The author must justify the validity of the semigroup property for Caputo fractional derivative.

The authors must justify the relation given by Eq.(57), because this relation is valid for $n$ an integer number (polynomial case) and $\alpha$ is not an integer number.

There are some misprints as below

  • Page 2, Line -16. Put a end point before the word Recently.
  • Page 3, Line 9. Insert the word "functions' between ...elementary -- is given..."
  • Page 3, Line -12. As you define Euler's integral transform with EIT, put EIT only.
  • Page 3, Line -8. As above.
  • Page 4, Line -3. As above.
  • Pag 6, Line 13. IV to VIII must be 5 to 7.
  • Page 12, Line -1. Insert the word `in' between ...functions -- terms...
  • Page 13, Line 2. Eq.(36) must be Eq.(42).
  • Page 14, Line -16. Insert the word "the" between ...show -- correspondence...
  • Page 14, Line -8. ...section 3 to 6 must be ...section 4 to 7.
  • Page 14, Line -5. Cancel the word "of".
  • Page 15, Line 1. Cancel the word "we" between the words have and shown.
  • Page 15, Eq.(58). It has two equal signs ++, cancel one of them.
  • Page 16, Line 12. As in Page 3, Line -12.
  • Page 17, Line -6. murray9s must be Murray's.
  • Page 17, Line -4. murray's must be Murray's.
  • Page 18, Line 5. murray must be Murray.
  • Page 18, Line 5. Page number?
  • Page 19, Line -3. press must be Press.

  • validity: good
  • significance: high
  • originality: high
  • clarity: high
  • formatting: good
  • grammar: excellent

Author:  Gavriil Shchedrin  on 2018-02-27  [id 220]

(in reply to Report 1 on 2018-02-15)

Warnings issued while processing user-supplied markup:

  • Inconsistency: plain/Markdown and reStructuredText syntaxes are mixed. Markdown will be used.
    Add "#coerce:reST" or "#coerce:plain" as the first line of your text to force reStructuredText or no markup.
    You may also contact the helpdesk if the formatting is incorrect and you are unable to edit your text.

Dear Editors,

We want to thank the Referee for reviewing our manuscript and providing valuable feedback. The Referee concluded that our manuscript provides ``an interesting connection between fractional derivatives and special functions, specifically some algebraic functions.'' The Referee requested clarification of the paper regarding three major points: the justification of the semigroup property for the Caputo fractional derivative, inclusion of a proof of the relation between the Hermite polynomials of non-integer order $\alpha$ and the Tricomi confluent hypergeometric function, and the resolution of the connection between the Caputo fractional derivative of integer order $\alpha$ and the regular integer order derivative. Following the Referee’s suggestions, we have modified the manuscript by addressing the raised points. Moreover, we have added an additional section on the exact evaluation of the Caputo derivative of a shifted polynomial and fixed numerous typos throughout the paper and references specified by the Referee.

As we have addressed all of the comments of the Referee, we hope our work will now be found suitable for SciPost. Please find below a detailed list of changes.

Sincerely, Gavriil Shchedrin, Nathanael C. Smith, Anastasia Gladkina, Lincoln D. Carr

{\itResponse to Referee}

  1. ``The definition given by Eq.(1) is valid for $0<\alpha<1$ with $\alpha\in{\mathbb{R}}$. The cases $\alpha=0$ and $\alpha=1$ must be treat as separated cases. See Kilbas, Srivastava and Trujillo, reference [1], page 92. This occurs also on pages 4, 7, 10, 11, 12, 13, specifically to obtain the graphics.''

Following the Referee’s suggestion we have included a paragraph right after Eq.(1) that now reads: ``Strictly speaking, the Caputo fractional derivative is given only for non-integer values of the fractional order, i.e., $\alpha\notin{\mathbb{N}}$ [1,2]. In the special case of the integer values of the parameter $\alpha=n\in{\mathbb{N}}$, the Caputo fractional derivative is {\it defined} in terms of the integer order derivative of the $n^{\rm th}$ order. We will find that except for the special case $\alpha=0$ the definition of the Caputo fractional derivative given by Eq.(1) for integer values of $\alpha=n\in{\mathbb{N}}$ is in fact consistent with the integer $n^{\rm th}$ order derivative. However, in the case of $\alpha=0$, the Caputo fractional derivative given by Eq.(1) results in a shift of the function by its value at the origin, i.e., ${}^{\rm C}D^{\alpha=0}_{x}f(x) = f(x)-f(0)$. Based on multiple examples of elementary functions ranging from trigonometric and inverse trigonometric functions to Gaussian and Lorentzian functions, we find that the Caputo fractional derivative of the order $0\leq{}\alpha\leq{1}$ {\it continuously} deforms the curve given by the Caputo fractional derivative of the $\alpha=0^{\rm th}$ order into $\alpha=1^{\rm st}$ order derivative. However, except for the special case $f(0)=0$, the requirement ${}^{\rm C}D^{\alpha=0}_{x}f(x) \equiv{} f(x)$ results in a discontinuous jump for the Caputo fractional derivative in the vicinity of $\alpha=0$. In order to avoid such a discontinuity, we impose the definition given by Eq.(1) for all $0\leq{}\alpha\leq{1}$.''

  1. ``The authors must justify the validity of the semigroup property for Caputo fractional derivative.''

Following the Referee’s suggestion we have included a paragraph at the end of the second section that now reads: `` The semi-group property for the Caputo fractional derivative of a factional order $0\leq{}\alpha\leq{1}$ can be directly established by expressing it in terms of the Riemann--Liouville fractional derivative,

$$ {}^{\rm C}D^{\alpha}{x}[f(x)] = {}^{\rm RL}D^{\alpha}[f(x) - f(0)] $$
Thus, the semi-group property of the {\Cp} derivative follows directly from the semi-group property of the Riemann--Liouville derivative [2]"

3.``The authors must justify the relation given by Eq.(57), because this relation is valid for n an integer number (polynomial case) and α is not an integer number."

Following the Referee’s suggestion we have included a paragraph at the end of Eq.(60) that now reads: `` The Hermite polynomials, $H_{\alpha}(x) $, of a positive non-integer order $\alpha$ are directly related to the parabolic cylinder function, $D_{\alpha}(z)$. This correspondence can be shown via the integral representation of the Hermite polynomials, [41,42,50,51]

$$ H_{\alpha}(x) = \frac{2^{\alpha + 1}}{\sqrt{\pi}}e^{x^{2}}\int^{\infty}{0} e^{-t^{2}}t^{\alpha}\cos\left(2xt - \frac{\pi\alpha}{2}\right)dt = \exp\left(\frac{x^2}{2}\right) 2^{\alpha /2} D x\right) . $$}\left(\sqrt{2
On the order hand, the parabolic cylinder function can be expressed in terms of the Tricomi confluent hypergeometric function [41,42,50,51], %
$$ D_{\alpha}(x) = \exp\left(-\frac{x^2}{4}\right) 2^{\alpha /2} U\left(-\frac{\alpha }{2},\frac{1}{2},\frac{x^2}{2}\right) . $$
Thus, the Hermite polynomials of non-integer order $\alpha$ are directly related to the Tricomi confluent hypergeometric function,
$$ H_{\alpha}(x) = 2^{\alpha}\,U\left(-\frac{\alpha}{2},\frac{1}{2},x^2\right). $$
"

  1. The Referee pointed out a number of misprints that have been promptly fixed.

  2. Finally, we have added section 8 where we report the Caputo fractional derivative of a shifted polynomial.

---

## Round 4 · Referee Report · Anonymous · 2018-4-6

Strengths

OK.

Weaknesses

OK.

Report

I agree with the publication.

Requested changes

Ok.

---

## Editorial Decision

published